# *O*-GlcNAcylation: An Emerging Protein Modification Regulating the Hippo Pathway

**DOI:** 10.3390/cancers14123013

**Published:** 2022-06-18

**Authors:** Eunah Kim, Jeong Gu Kang, Eek-hoon Jho, Won Ho Yang, Jin Won Cho

**Affiliations:** 1Glycosylation Network Research Center, Yonsei University, 50 Yonsei-ro, Seodaemun-gu, Seoul 03722, Korea; bioniceunah@yonsei.ac.kr (E.K.); ej70@uos.ac.kr (E.-h.J.); bionicwono@yonsei.ac.kr (W.H.Y.); 2Genome Editing Research Center, Korea Research Institute of Bioscience & Biotechnology (KRIBB), 125 Gwahak-ro, Yuseong-gu, Daejeon 34141, Korea; kang@kribb.re.kr; 3Department of Bio-Molecular Science, KRIBB School of Bioscience, Korea University of Science and Technology (UST), Daejeon 34141, Korea; 4Department of Life Science, University of Seoul, 163 Seoulsiripdae-ro, Dongdaemun-gu, Seoul 02504, Korea; 5Department of Systems Biology, College of Life Science and Biotechnology, Yonsei University, 50 Yonsei-ro, Seodaemun-gu, Seoul 03722, Korea

**Keywords:** Hippo pathway, *O*-GlcNAcylation, cancer, cellular signaling pathway

## Abstract

**Simple Summary:**

The contact point between the Hippo pathway, which serves as a central hub for various external environments, and *O*-GlcNAcylation, which is a non-canonical glycosylation process acting as a dynamic regulator in various signal transduction pathways, has recently been identified. This review aims to summarize the function of *O*-GlcNAcylation as an intrinsic and extrinsic regulator of the Hippo pathway.

**Abstract:**

The balance between cellular proliferation and apoptosis and the regulation of cell differentiation must be established to maintain tissue homeostasis. These cellular responses involve the kinase cascade-mediated Hippo pathway as a crucial regulator. Hence, Hippo pathway dysregulation is implicated in diverse diseases, including cancer. *O*-GlcNAcylation is a non-canonical glycosylation that affects multiple signaling pathways through its interplay with phosphorylation in the nucleus and cytoplasm. An abnormal increase in the *O*-GlcNAcylation levels in various cancer cells is a potent factor in Hippo pathway dysregulation. Intriguingly, Hippo pathway dysregulation also disrupts *O*-GlcNAc homeostasis, leading to a persistent elevation of *O*-GlcNAcylation levels, which is potentially pathogenic in several diseases. Therefore, *O*-GlcNAcylation is gaining attention as a protein modification that regulates the Hippo pathway. This review presents a framework on how *O*-GlcNAcylation regulates the Hippo pathway and forms a self-perpetuating cycle with it. The pathological significance of this self-perpetuating cycle and clinical strategies for targeting *O*-GlcNAcylation that causes Hippo pathway dysregulation are also discussed.

## 1. Introduction

The evolutionarily conserved Hippo pathway influences tissue growth and development by regulating proliferation, apoptosis, and cell differentiation [1,2,3]. Its dysregulation can trigger tumorigenesis, tissue fibrosis, and hyperplasia [4,5]. The central components of the Hippo pathway are serine/threonine kinases and effectors, namely, the Yes-associated protein (YAP) and the transcriptional co-activator with PDZ-binding motif (TAZ) [4,6,7,8]. Thus, the Hippo pathway is driven by kinase cascade-mediated phosphorylation. Since 2017, when *O*-GlcNAcylation was found to be a posttranslational modification (PTM) regulating the Hippo pathway, it has been in the spotlight in the Hippo pathway research field [9,10,11,12,13]. *O*-GlcNAcylation modulates various phosphorylation-mediated signaling pathways via crosstalk [14,15]. In tumors from multiple tissues, abnormal increases in *O*-GlcNAc transferase (OGT), an enzyme catalyzing *O*-GlcNAcylation, and intracellular *O*-GlcNAcylation have been observed along with hyperactivation of YAP/TAZ [12,16,17,18,19,20,21,22]. Moreover, some core components of the Hippo pathway are *O*-GlcNAcylated, and these *O*-GlcNAcylations induce Hippo pathway dysregulation [9,10,12]. In addition, when Hippo pathway dysregulation causes hyperactivated YAP, the OGT expression increases, leading to consistently high intracellular *O*-GlcNAcylation levels [4,9,10]. Aberrant hyper-*O*-GlcNAcylation and Hippo pathway dysregulation are implicated in tumorigenesis, tumor growth, and metastasis, and they contribute to various diseases [4,5,20,23,24,25,26]. Thus, understanding how *O*-GlcNAcylation affects Hippo pathway dysregulation is necessary to end the vicious cycle caused by the feedback regulation between *O*-GlcNAcylation and the Hippo pathway.

This review provides overviews of the Hippo pathway and *O*-GlcNAcylation and suggests mechanisms through which *O*-GlcNAcylation induces Hippo pathway dysregulation by examining the *O*-GlcNAcylation of specific proteins that regulate the Hippo pathway. In addition, this review presents the pathological significance of the self-perpetuating cycle caused by the feedback between *O*-GlcNAcylation and the Hippo pathway. It also discusses the applicability and limitations of clinical techniques to break such a mutual relationship.

## 2. Hippo Pathway

The Hippo pathway is an evolutionarily conserved intracellular signaling pathway that affects tissue growth and development by coordinating apoptosis, cell proliferation, and cell differentiation [1,2,3]. It was first identified through the genetic screening of Drosophila melanogaster to discover tissue growth regulators [1,6,27,28,29,30,31,32,33]. Since the central components of the Hippo pathway were identified in mammalian cells, numerous functional studies on the Hippo pathway in mammalian cells have been conducted.

### 2.1. Kinase-Mediated Signaling Cascade of the Hippo Pathway

The central part of the mammalian Hippo pathway is a kinase cascade consisting of the serine/threonine kinases mammalian Ste-20-like kinase 1/2 (MST1/2) and large tumor suppressor 1/2 (LATS1/2), their adaptor proteins Salvador homologue 1 (SAV1) and Mps one binder 1 (MOB1), and the transcriptional co-activators YAP and TAZ (Figure 1). When MST1/2 are phosphorylated at the activation loop (Thr183/Thr180) by specific stimuli, MST1/2 activation is induced by phosphorylation-mediated conformational change [34]. The direct binding between SAV1 and MST1/2 facilitates MST1/2 activation by promoting and protecting the phosphorylation of the MST1/2 activation loop and stabilizing SAV1 [35,36]. SAV1 also recruits MST1/2 to the plasma membrane [37,38]. Activated MST1/2 interacts with MOB1 by creating a phosphor-docking site via autophosphorylation, causing a conformational change in MOB1 and allowing MOB1 to bind to LATS1/2 [39,40]. Merlin recruits LATS1/2 to the MST1/2-SAV1 complex at the plasma membrane [38]. As a result, MST1/2 can indirectly interact with LATS1/2 with the assistance of Merlin and two adaptors, namely SAV1 and MOB1 [38,39,40]. MST1/2 phosphorylate the N-terminal tail of MOB1 (Thr12 and Thr35) and the hydrophobic motif of LATS1/2 (Thr1079/Thr1041) [40]. MOB1 phosphorylation causes the LATS1/2–MOB1 complex to dissociate from the MST1/2–SAV complex, and the released LATS1/2 are activated by the autophosphorylation of its activation loop (Ser909/Ser872) [40,41]. Activated LATS1/2 subsequently phosphorylate a serine residue within the HXRXXS motifs of the Hippo pathway effectors YAP/TAZ [8,42,43,44,45]. The phosphorylated residues of YAP are Ser61, Ser109, Ser127, Ser164, and Ser381, and those of TAZ are Ser66, Ser89, Ser117, and Ser311. When YAP/TAZ are phosphorylated at Ser127/Ser89, an interaction with 14-3-3 occurs, inducing the cytoplasmic sequestration of YAP and TAZ [8,42,43]. When YAP/TAZ are also phosphorylated at Ser381/Ser311, phosphorylation by casein kinase 1δ/ε (CK1δ/ε) is stimulated, causing the proteasomal degradation of YAP and TAZ via ubiquitination by SCF (SKP1 (S-phase-kinase-associated protein 1), Cullin-1, and F-box protein) E3 ubiquitin ligase [44,45]. Therefore, phosphorylated YAP/TAZ cannot act as transcriptional cofactors [8], whereas dephosphorylated YAP/TAZ interact with and assist the transcription factors, such as TEADs, p73, RUNX, and TBX5 [46,47,48,49,50,51].

### 2.2. Hippo Pathway as an Essential Cellular Hub

The Hippo pathway is controlled by diverse mechanical or chemical upstream inputs rather than exclusive molecules (Figure 1). Cell adhesion enhances Hippo signaling by attracting MST1/2 and LATS1/2 closer to each other or sequestering YAP/TAZ at cell junctions [52]. Mechanical cues regulate the Hippo pathway by actin remodeling, and extracellular signaling molecules such as hormones, growth factors, and lysophosphatidic acid (LPA) modulate the Hippo pathway via their receptors or G protein-coupled receptors (GPCRs) [52,53]. The Hippo pathway is also controlled by stresses such as glucose deprivation [54,55,56], hypoxia [57], endoplasmic reticulum stress [55], heat shock [58,59,60], osmotic stress [61], and oxidative stress [62,63]. Recently, the striatin-interacting phosphatase and kinase (STRIPAK) complex was reported as an upstream regulator of the Hippo pathway. It dephosphorylates and suppresses MST1 in a manner dependent on Ras homolog family member A (RhoA) [64]. The STRIPAK mechanism can explain how upstream signals such as LPA and serum connect to one another to regulate the Hippo signaling pathway, but their correlation with other stimuli needs to be further investigated.

The Hippo pathway is regulated by numerous signaling pathways [65], namely the Wnt [66], Ras–Raf–MAPK [67], TGFβ [68,69], Hedgehog [70], Notch [71,72,73,74], MAPK [75,76], and Mevalonate pathways [77]. It acts as an essential hub for transmitting diverse inputs to YAP/TAZ and transducing such signals into cellular responses, including proliferation, survival, metastasis, stemness, and regeneration [46,47,78]. Hence, the activity of YAP/TAZ must be fine-tuned. YAP/TAZ modulate their own activities by inducing the gene expression of Merlin and LATS2, which act as negative regulators of YAP/TAZ [79,80]. This negative feedback loop in the Hippo pathway helps establish YAP/TAZ homeostasis. The hyperactivation of YAP and TAZ via Hippo pathway dysregulation leads to tumorigenesis, tissue fibrosis, and hyperplasia in some organs; promotes tumor growth and metastasis; and confers chemotherapeutic resistance to some cancer cells [5,81]. Indeed, Hippo pathway dysregulation is frequently observed in patients with cancer or fibrotic diseases [5].

## 3. *O*-GlcNAcylation

YAP/TAZ hyperactivation is a common feature of cancer cells, but the genetic mutations of the core components in the Hippo pathway are rarely found in patients with cancer [4,16,82,83,84]. This observation begs the question of what causes Hippo pathway dysregulation in cancer cells? The abnormal elevation of intracellular *O*-GlcNAcylation in cancer cells is one possible answer.

### 3.1. Hexosamine Biosynthetic Pathway

*O*-GlcNAcylation is a PTM through which a single N-acetylglucosamine (GlcNAc) is attached to a target protein [85,86]. Uridine diphosphate-N-acetylglucosamine (UDP-GlcNAc), an active monosaccharide donor, is synthesized by the hexosamine biosynthetic pathway (HBP) that consolidates glucose, amino acid, fatty acid, and nucleotide metabolism [87] (Figure 2). Thus, *O*-GlcNAcylation acts as a nutrient sensor. Glucose, which is transported into cells through glucose transporters (GLUT), is phosphorylated by hexokinase using ATP, thereby producing glucose-6-phosphate (Glc-6-p); Glc-6-p is then transformed into fructose-6-phosphate (Fru-6-P). Most of these products undergo pentose phosphate and glycolytic pathways as energy and carbon sources. A small portion of Fru-6-P is turned into glutamine-fructose-6-phosphate (GlcN-6-P) by glutamine fructose-6-phosphate amidotransferase (GFAT), the key enzyme of the HBP. Thus, only 2–5% of imported glucose can be converted to UDP-GlcNAc using glutamine, acetyl-CoA, and uridine-5′-triphosphate (UTP) through the HBP [87,88,89]. The GlcNAc moiety is transferred by OGT from UDP-GlcNAc to the specific serine/threonine residues of various target proteins [90,91]. *O*-GlcNAc from modified proteins is hydrolyzed by *O*-GlcNAcase (OGA) [92].

### 3.2. OGT and OGA: The Sole Enzymes Responsible for the Intracellular O-GlcNAcylation Cycle

*O*-GlcNAc modifications of thousands of intracellular proteins are reversibly and dynamically regulated by two enzymes, namely OGT and OGA [85]. OGT is an exclusive enzyme involved in *O*-GlcNAcylation [90,93]. Human OGT (hOGT) contains 2.5–13.5 tetratricopeptide repeats (TPRs), a linker domain, and C-terminal catalytic domains [94,95]. Three hOGT variants are derived by alternative splicing and multiple transcription start sites (Figure 3A). Among them, nucleocytoplasmic OGT (ncOGT) is the longest (with 13.5 TPR repeats) and most abundant OGT variant. The shortest OGT (sOGT) possesses 2.5 TPR repeats. ncOGT and sOGT are both found in the nucleus and cytoplasm. Mitochondrial OGT (mOGT) contains nine TPR repeats, and its location is due to a mitochondrial targeting sequence (MTS) in the N-terminal region. The substrate recognition of OGT relies on TPR repeats [96]. These conserved tandem repeats of 34 amino acids form a superhelix structure, and the asparagine ladder in the superhelix mediates the recognition between OGT and its diverse substrates [97]. Moreover, diverse adapter proteins that recruit OGT to specific substrates depending on the cellular conditions confer OGT substrate selectivity and substrate diversity [98].

Similar to OGT, OGA recognizes diverse substrates. Human OGA has two distinct splice variants (Figure 3B): nucleocytoplasmic OGA (ncOGA), which is located primarily in the cytoplasm, and short OGA (sOGA), which is located primarily in the nucleus and lipid droplets [99,100]. Both OGA variants possess an N-terminal hydrolase catalytic domain that hydrolyzes *O*-GlcNAc modifications. However, sOGA does not have the C-terminal histone acetyltransferase (HAT)-like domain and part of the stalk domain. The stalk domain participates in forming an OGA homodimer in which a potential substrate-binding cleft is created by covering the catalytic domain of the sister monomer OGA [101,102,103]. Through this substrate-binding cleft, binding to a GlcNAc moiety and sequence-independent peptide backbone interactions with the substrate are possible [101,102,103]. In addition, sequence-dependent side chain interactions can occur within the substrate-binding cleft [104]. Hence, interactions within the OGA substrate-binding cleft likely allow OGA to differentially regulate the *O*-GlcNAcylation turnover rate for various substrates [104]. Due to the lack of these interactions in sOGA, the hydrolase activity of sOGA is much weaker than that of ncOGA. Human OGA was expected to possess histone acetyltransferase activity due to the similarity of its HAT-like domain to GCN5-related N-acetyltransferase (GNAT) [105]. However, the P-loop motif that supports acetyl-CoA binding is absent from the HAT-like domain of hOGA [106,107]. Thus, the HAT-like domain of OGA is a pseudo-HAT, but its function remains unclear. Although these structural studies have provided insights into the interactions of OGA with various substrates, further studies are needed to explain clearly how OGA is regulated to recognize substrates and investigate the functional roles of the HAT-like domain in substrate recognition by OGA.

*O*-GlcNAcylation affects the cellular processes involved in gene expression and signal transduction by regulating chromatin remodeling and protein stability, activity, localization, and protein–protein interactions [98,108,109]. In particular, it plays a role in many cellular signaling pathways through its reciprocal effects with phosphorylation [14,15]. Thus, *O*-GlcNAc homeostasis must be maintained within cells to sustain normal cellular functions, and it is accomplished by OGT and OGA [98]. OGT and OGA are mutually regulated in terms of the gene transcription levels, protein activity, and protein stability [98]. The *O*-GlcNAcylation of OGT and OGA is also thought to play a role in maintaining *O*-GlcNAc homeostasis [98]. OGT *O*-GlcNAcylation decreases the overall level of intracytoplasmic *O*-GlcNAcylation (Figure 3C). OGT *O*-GlcNAcylation at Ser389 promotes the nuclear import of OGT by facilitating its interaction with importin α5 [110]. OGT *O*-GlcNAcylation at Ser3 and Ser4 also decreases OGT activity by competing with GSK3β-mediated phosphorylation, which enhances the OGT activity [111]. However, OGA *O*-GlcNAcylation at Ser405 reduces its stability and enzymatic activity [112,113] (Figure 3D). Hence, additional research is needed to support the hypothesis that the *O*-GlcNAcylation of OGA is involved in maintaining *O*-GlcNAc homeostasis.

## 4. Effect of *O*-GlcNAcylation on the Hippo Pathway

Glucose is a major factor regulating the Hippo pathway. As a representative energy sensor, AMP-activated protein kinase (AMPK) activated in response to glucose deprivation induces YAP phosphorylation in a LATS-dependent and LATS-independent manner; consequently, it interferes with the binding of YAP to TEAD [54,55]. Glucose metabolism enhances YAP/TAZ transcriptional activity. Phosphofructokinase 1 (PFK1), the key enzyme in the first step of glycolysis, binds to TEAD and functionally cooperates with YAP/TAZ [114]. Glucose metabolism and the Hippo pathway are also connected by *O*-GlcNAcylation synthesized by the HBP, which branches from glycolysis. YAP activity is enhanced by an increase in cellular *O*-GlcNAcylation levels via OGT overexpression or treatment with PUGNAc, an OGA inhibitor [9,10]; conversely, such activity is attenuated by a decrease in *O*-GlcNAcylation via OGT knockdown or treatment with OSMI, an OGT inhibitor [9,12].

Cancer cells enhance the glucose uptake to meet the increased energy and metabolism demands for cell growth and proliferation [115,116]. In cancer cells, excessive glucose uptake and increased GFAT, the rate-limiting enzyme in the HBP, cause an increase in UDP-GlcNAc from the HBP flux [117,118]. Together with an increase in UDP-GlcNAc synthesis, OGT overexpression in cancer cells causes aberrant hyper-*O*-GlcNAcylation [21,22,23]. *O*-GlcNAcylation enhances YAP/TAZ activity and YAP/TAZ induce an increase in the cellular *O*-GlcNAcylation levels [9,10,12,79,80]. This mutual relationship drives a self-perpetuating cycle that sustains aberrant hyper-*O*-GlcNAcylation and Hippo pathway dysregulation [12]. Hence, *O*-GlcNAcylation should be studied as a factor regulating the Hippo pathway. In this section, we describe *O*-GlcNAcylation associated with Hippo pathway dysregulation and suggest potential mechanisms through which *O*-GlcNAcylation affects the Hippo pathway by integrating the results of studies on *O*-GlcNAcylation in intracellular signaling pathways that crosstalk with the Hippo pathway.

### 4.1. Mechanism by Which O-GlcNAcylation Induces Hippo Pathway Dysregulation

Since YAP *O*-GlcNAcylation was reported in 2017, several studies have been conducted on the *O*-GlcNAcylation of core components in the Hippo pathway kinase cascade [9,10,11,12]. Subsequently, the *O*-GlcNAcylation of MST1 and LATS2 has been confirmed [12]. However, the *O*-GlcNAcylation of MST2 and LATS1, whose sequences are similar to those of MST1 and LATS2, has not been detected [9,12]. Furthermore, the *O*-GlcNAcylation of TAZ, SAV, and MOB has not been observed [9,11] Although the effect of MST *O*-GlcNAcylation on the Hippo pathway is unclear, the *O*-GlcNAcylation of YAP and LATS2 is closely associated with Hippo pathway dysregulation (Figure 4). The *O*-GlcNAcylation of LATS2 at Thr436 interferes with the interaction between LATS2 and MOB1, decreasing the LATS2 activity by inhibiting MST-mediated phosphorylation [12]. Thus, the *O*-GlcNAcylation of LATS2 increases the activity of YAP/TAZ [12]. The *O*-GlcNAcylation of YAP at Ser109 or Thr241 also enhances the activity of YAP by inhibiting its interaction with LATS1 [9,10].

In addition to the *O*-GlcNAcylation of the core components in the Hippo kinase cascade, the *O*-GlcNAcylation of angiomotin (AMOT) and LDL receptor-related protein 6 (LRP6) is possibly implicated in Hippo pathway dysregulation [11,13] (Figure 4). AMOT affects cancer growth and invasion via several signaling pathways: mTOR, MAPK, Wnt signaling, and the Hippo pathway [119]. However, studies have yet to verify whether AMOT acts as an oncoprotein or a tumor suppressor, because its effect on tumor growth differs depending on the cancer cell type [119]. For example, the effect of AMOT on the Hippo pathway varies depending on the cell type. AMOT acts as an oncoprotein by increasing the activity of YAP in hepatic carcinoma, but it acts as a tumor suppressor by repressing the activation of YAP target genes in ovarian cancer [119]. AMOT has two isoforms, namely AMOT-p130 and AMOT-p80, due to alternative splicing. AMOT-p130, which can interact with YAP via PPxY motifs in its N-terminal region, undergoes *O*-GlcNAcylation [11,120]. In liver cancer cells, the effect of AMOT on the Hippo pathway depends on the concentration of glucose, a major source of UDP-GlcNAc. AMOT functions as a YAP suppressor under normal glucose conditions, but under high glucose conditions, AMOT induces the nuclear accumulation of YAP, thereby enhancing the pro-tumorigenic function of YAP [11]. LRP6, a co-receptor of canonical Wnt signaling, also affects the Hippo pathway by binding to Merlin [121,122]. This interaction suppresses the Hippo pathway by reducing the interaction between Merlin and LATS1/2 [13,121]. Under nutrient starvation conditions, such as serum- or glucose-free culture, LRP6 *O*-GlcNAcylation decreases, and the endocytosis-mediated lysosomal degradation of LRP6 increases. As a result, more Merlin becomes available to interact with LATS1/2, and the YAP activity decreases [13]. These results further support that *O*-GlcNAcylation indirectly attenuates the Hippo pathway. Further research that identifies *O*-GlcNAcylation sites in AMOT and LRP6 is needed to elucidate the function of AMOT and LRP6 *O*-GlcNAcylation.

Excessive YAP/TAZ activation can be prevented, because YAP triggers the transcription of Merlin and LATS2, which are negative regulators of YAP and TAZ [79,80]. However, this negative feedback loop can be blocked by LATS2 *O*-GlcNAcylation and, even if more LATS2 is recruited to the MST-MOB1 complex by an increase in Merlin and LATS2 transcription, LATS2 *O*-GlcNAcylation inhibits the interaction between the MST-MOB1 complex and LATS2 [12]. Hence, abnormally increased *O*-GlcNAcylation can disrupt Hippo pathway homeostasis, leading to persistent YAP and TAZ hyperactivation. Interestingly, activated YAP also promotes glucose uptake by enhancing the GLUT3 gene expression and increases HBP-stimulating intracellular metabolites, such as glutamine, acetyl-CoA, and Fru-6-P [10,54]. In addition, YAP enhances OGT transcription, which increases the overall intracellular *O*-GlcNAcylation levels [9,10]. In summary, aberrantly increased *O*-GlcNAcylation induces a positive feedback loop that sustains a hyper-*O*-GlcNAcylation state via Hippo pathway dysregulation. These findings imply that increased *O*-GlcNAcylation triggers Hippo pathway dysregulation in cancer cells and maintains a hyper-*O*-GlcNAcylation state, leading to tumor growth and metastasis. In xenograft mouse experiments that observe the effects of YAP and LATS2 *O*-GlcNAcylation on tumor growth, tumors from grafts expressing an *O*-GlcNAcylation-deficient YAP (S109A or T241A) or LATS2 (T436A) mutant are significantly smaller than those from grafts expressing wild-type YAP or LATS2 [9,10,12].

### 4.2. O-GlcNAcylation in Cellular Signaling Pathways That Crosstalk with the Hippo Pathway

The Hippo pathway crosstalks with multiple cellular signaling pathways, such as Wnt, TGFβ, GPCR, and Notch [65]. Because some of these signaling pathways are regulated by *O*-GlcNAcylation, the Hippo pathway is expected to be indirectly affected by *O*-GlcNAcylation (Figure 5). For example, β-catenin, an essential mediator of the canonical Wnt signaling pathway, is *O*-GlcNAcylated at Thr41 in its N-terminus [123]. This *O*-GlcNAcylation increases β-catenin stability by competing with ubiquitination-inducing phosphorylation that occurs in the absence of a Wnt activity [123]. Activated β-catenin induces the transcriptional upregulation of YAP by forming a β-catenin/TCF4 complex that binds to a DNA enhancer element within YAP in colorectal cancer cells [124]. Phosphorylated β-catenin also induces the proteasomal degradation of TAZ by bridging TAZ to β-TrCP, a ubiquitin ligase [66]. Hence, *O*-GlcNAcylation may indirectly enhance YAP/TAZ activity by controlling Wnt signaling. Likewise, the *O*-GlcNAcylation of Smad4, an important regulator of the TGFβ signaling pathway, at Thr63 prevents the GSK3β-mediated proteasomal degradation of Smad4, inducing the TGFβ signaling pathway [125]. SnoN, a target gene of TGFβ signaling, stabilizes TAZ by preventing phosphorylation by LATS [68]. Thus, Smad4 *O*-GlcNAcylation likely promotes YAP/TAZ activity by inducing TGFβ signaling. However, the *O*-GlcNAcylation of PKC decreases the TGFβRII expression by diminishing PKC activity; as a result, TGFβ signaling is reduced [126,127]. Therefore, the effect of *O*-GlcNAcylation on the Hippo pathway via TGFβ signaling may vary depending on OGT target proteins. PKA, a protein kinase that bridges the Hippo pathway and GPCR-Gαs signaling by enhancing LATS1/2 activity through the direct phosphorylation of LATS1/2 or the suppression of actin fiber formation, is also *O*-GlcNAcylated [128,129,130]; consequently, PKA kinase activities are enhanced [131]. Therefore, the *O*-GlcNAcylation of PKA may enhance the Hippo pathway through the GPCR signaling pathway. NOTCH1 *O*-GlcNAcylation induces the release of the Notch intracellular domain (NICD) by enhancing DLL1-NOTCH and DLL4-NOTCH1 interaction [132]. NICD promotes YAP/TAZ stability, thereby enhancing YAP/TAZ activity [71,72,73]. Hence, *O*-GlcNAcylation may improve YAP/TAZ stability by regulating NOTCH signaling. Collectively, these studies imply that *O*-GlcNAcylation indirectly affects the Hippo pathway by regulating its associated pathways. However, these conclusions are derived by integrating individual findings from multiple studies; thus, confirmatory studies are needed. Table 1 summarizes the *O*-GlcNAc proteins involved in the Hippo pathway and the action mode of *O*-GlcNAcylation.

## 5. Conclusions

Since the discovery of the Hippo pathway in the early 21st century, its phosphorylation-mediated signaling has been elucidated. Although phosphorylation is the primary mechanism of Hippo pathway regulation, it is affected by several PTMs, such as ubiquitination, acetylation, methylation, sumoylation, and *O*-GlcNAcylation [9,10,11,12,133]. Particularly, *O*-GlcNAcylation, which can crosstalk with phosphorylation, causes Hippo pathway dysregulation, leading to continuous YAP/TAZ hyperactivation. In addition, hyperactivated YAP increases intracellular glucose and HBP-stimulated metabolite concentrations and promotes OGT gene expression, which abnormally increases intracellular *O*-GlcNAcylation. This mutual relationship between *O*-GlcNAcylation and the Hippo pathway causes a self-perpetuating cycle that disrupts intracellular *O*-GlcNAc homeostasis, thereby sustaining aberrant hyper-*O*-GlcNAcylation and Hippo pathway dysregulation. Hippo pathway dysregulation and aberrant increases in intracellular *O*-GlcNAcylation have been observed in cancer cells derived from various tissues, and they contribute to carcinogenesis and cancer progression. Thus, Hippo pathway components and *O*-GlcNAcylation regulatory enzymes (OGT and OGA) are potential targets for cancer diagnosis and treatment. Currently, Hippo pathway-targeting compounds, such as Verteporfin, and various OGT- and OGA-targeting molecular probes, such as OSMI-1 and Thiamet-G, are being developed. However, the systemic application of these compounds causes severe side effects because OGT and OGA exclusively control the *O*-GlcNAcylation of numerous vital intracellular proteins and the Hippo pathway is also involved in tissue and organ growth, development, regeneration, repair, and immune modulation. With the development of compounds targeting specific *O*-GlcNAcylation that induces Hippo pathway dysregulation, new cancer treatment approaches can be established. Such compounds can be developed into a wide range of medical applications due to the diversity of diseases associated with *O*-GlcNAc homeostasis disruption and Hippo pathway dysregulation, including inappropriate immune responses, excessive fibrosis, and metabolic disorders. YAP *O*-GlcNAcylation disturbs YAP-LATS1 interactions; LATS2 *O*-GlcNAcylation enhances YAP/TAZ activity and stability and blocks the negative feedback loop of the Hippo pathway, resulting in persistent YAP/TAZ hyperactivation. Hence, YAP *O*-GlcNAcylation and LATS2 *O*-GlcNAcylation are excellent target candidates.

Our understanding of how OGT and OGA select target proteins differently depending on the cellular environment is insufficient, and the technology that targets only the *O*-GlcNAcylation of specific target proteins is not secure. Moreover, the fundamental biology underlying the interactions between *O*-GlcNAcylation and the Hippo pathway needs additional research. With additional knowledge about the mutual relationship between *O*-GlcNAcylation and the Hippo pathway and the development of techniques for detecting and regulating the *O*-GlcNAcylation of specific target proteins, more therapeutics and regenerative medicine products can be discovered to cure human diseases.

## Figures and Tables

**Figure 1 cancers-14-03013-f001:**
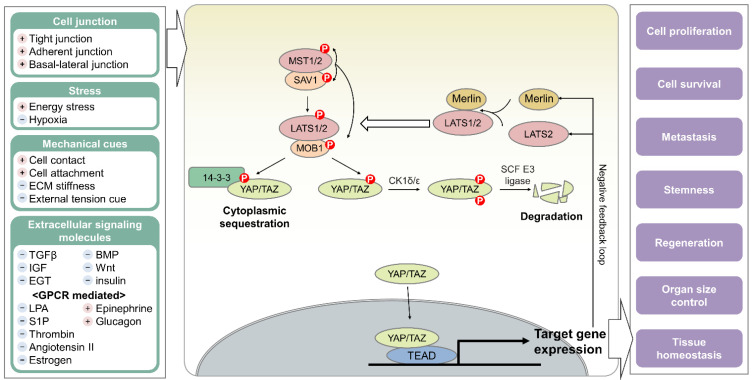
Schematic representation of the mammalian Hippo pathway. The Hippo kinase cascade, composed of Ser/Thr kinases MST1/2 and LATS1/2, adaptor proteins SAV1 and MOB1, and effectors YAP/TAZ, is regulated by various stimuli, including cell–cell junctions, cellular stresses, mechanical cues, and multiple extracellular signaling molecules. A “+” indicates a stimulus that increases the activity of the Hippo pathway, and a “−” indicates a stimulus that decreases activity. The Hippo pathway phosphorylates YAP/TAZ, leading to their cytoplasmic sequestration and proteasomal degradation. Dephosphorylated YAP/TAZ translocate into the nucleus and act as transcriptional cofactors, thereby controlling cellular responses such as proliferation, survival, and metastasis and affecting stemness, regeneration, organ size, and tissue homeostasis.

**Figure 2 cancers-14-03013-f002:**
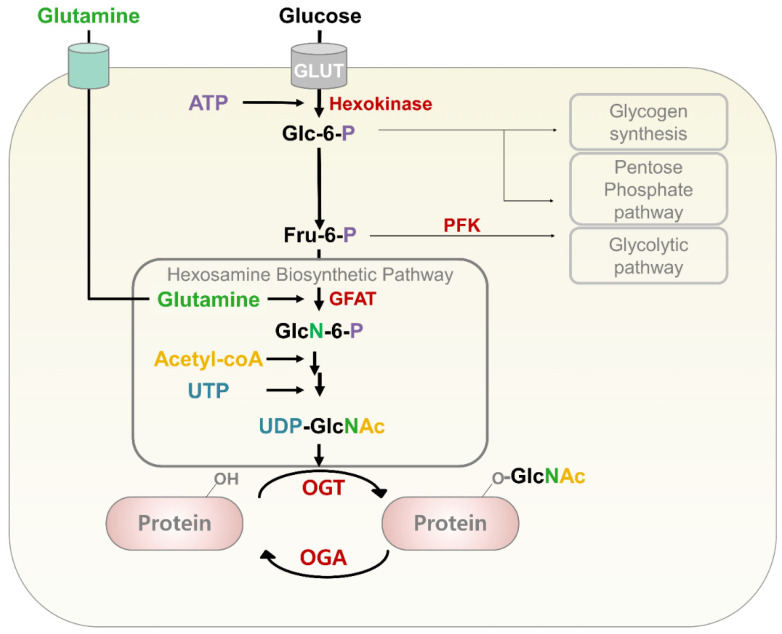
UDP-GlcNAc synthesis via the HBP and *O*-GlcNAc cycling. In the HBP, UDP-GlcNAc, an active monosaccharide donor for *O*-GlcNAcylation, is synthesized by consolidating glucose, glutamine, acetyl-CoA, and UTP, which are metabolites of carbohydrates, proteins, lipid acids, and nucleotides, respectively. The GlcNAc moiety of UDP-GlcNAc is transferred by OGT to the hydroxyl group of Ser/Thr residues on target proteins. *O*-GlcNAc from target proteins is then hydrolyzed by OGA.

**Figure 3 cancers-14-03013-f003:**
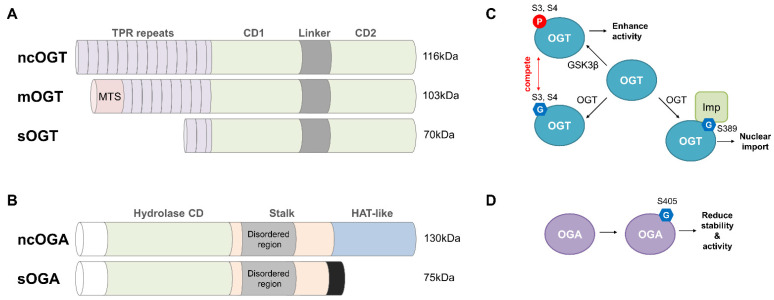
Exclusive enzymes directly involved in the *O*-GlcNAcylation cycle. (**A**) The schematic structure of hOGT isoforms. Three hOGT variants (ncOGT, mOGT, and sOGT) are derived from the OGT gene located on chromosome Xq13.1 by alternative splicing and multiple transcription start sites. These variants possess an identical catalytic domain in the C-terminal region, but they have different TPR repeats involved in substrate recognition in the N-terminal region. ncOGT (116 kDa) has 13.5 TPR repeats, mOGT (103 kDa) contains 9 TPR repeats, and sOGT possesses only 2.5 TPR repeats. Additionally, only mOGT contains a mitochondrial targeting sequence (MTS) in the N-terminal region. (**B**) The structure of human OGA (hOGA) isoforms. Two hOGA variants (ncOGA and sOGA) are produced from the MGEA5 gene located on chromosome 10q24.32. Both contain a hydrolase catalytic domain in the N-terminal region, but only ncOGA has a HAT-like domain and part of the stalk domain. (**C**) The function of OGT *O*-GlcNAcylation. *O*-GlcNAcylation at Ser389 of OGT promotes interaction with importin α5, leading to the nuclear import of OGT. *O*-GlcNAcylation at Ser3 and Ser4 of OGT inhibits its activity by competing with GSK3β-mediated phosphorylation. (**D**) The function of OGA *O*-GlcNAcylation. *O*-GlcNAcylation at Ser405 of OGA represses its stability and activity.

**Figure 4 cancers-14-03013-f004:**
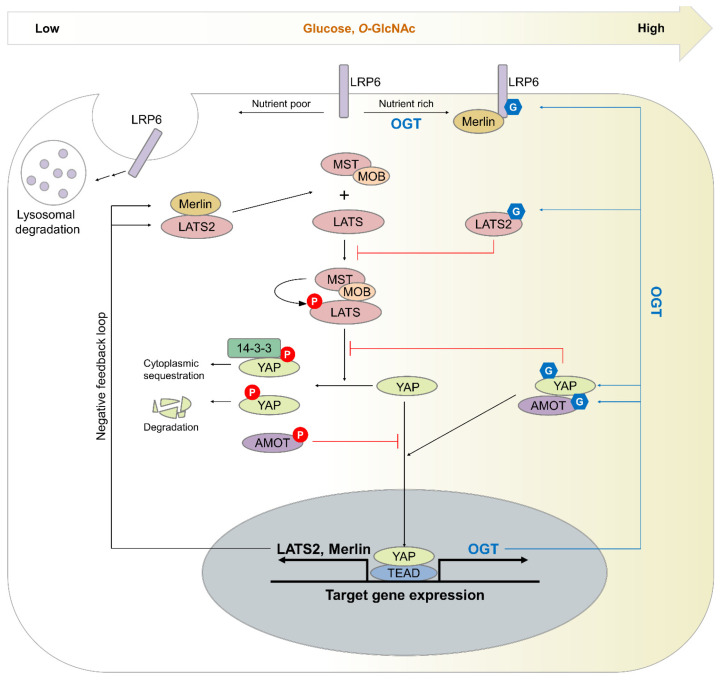
Mechanism by which *O*-GlcNAcylation constantly attenuates the Hippo pathway through mutual regulation. The *O*-GlcNAcylation of specific proteins, such as LRP6, LATS2, YAP, and AMOT, increases the activity of the Hippo pathway effector YAP. LRP6 *O*-GlcNAcylation may diminish LATS activity by decreasing Merlin–LATS interactions through the inhibition of the lysosomal degradation of LRP6. LATS2 *O*-GlcNAcylation inhibits its activity by interrupting the MOB1–LATS2 interaction. YAP *O*-GlcNAcylation induces its activity by disturbing the interaction with LATS1. AMOT *O*-GlcNAcylation may cause the nuclear accumulation of YAP by decreasing AMOT phosphorylation at Ser175. Hyperactivated YAP induces the gene expression of LATS2, Merlin, and OGT. LATS2 *O*-GlcNAcylation blocks the Hippo pathway negative feedback loop caused by YAP-mediated LATS2/Merlin gene expression by blocking MOB1-LATS2 interactions. As a result, abnormally increased *O*-GlcNAcylation induces Hippo pathway dysregulation and sustains aberrant hyper-*O*-GlcNAcylation.

**Figure 5 cancers-14-03013-f005:**
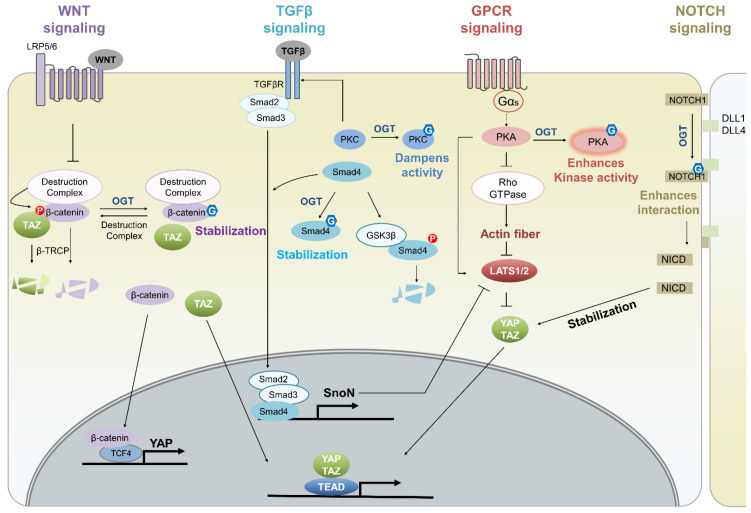
The schematic model of *O*-GlcNAcylation regulating the Hippo pathway through other pathways that crosstalk with the Hippo pathway. The *O*-GlcNAcylation of β-catenin, a mediator of the canonical Wnt signaling pathway, competes with ubiquitinylation-inducing β-catenin phosphorylation, which may stabilize β-catenin and TAZ and promote β-catenin/TCF4 complex-mediated YAP expression. Smad4 *O*-GlcNAcylation enhances the TGF-β/SMAD signaling pathway, which can upregulate SnoN gene expression, thereby inactivating LATS1/2 by stabilizing Smad4. PKC *O*-GlcNAcylation is possibly related to TGF-β signaling in a way that TGFβRII expression is decreased by reducing PKC activities. In Gαs-coupled GPCR signaling, PKA *O*-GlcNAcylation may enhance LATS1/2 activity by increasing PKA-mediated LATS1/2 phosphorylation or inhibiting actin fiber formation. NOTCH1 *O*-GlcNAcylation elicits the release of NICD, which can stabilize YAP/TAZ, by enhancing the interaction between NOTCH1 and DLL1 or DLL4.

**Table 1 cancers-14-03013-t001:** Summary of the *O*-GlcNAc proteins involved in the Hippo pathway and the action mode of *O*-GlcNAcylation.

Protein	*O*-GlcNAc Site	Targeted Pathway	Function	Refs
YAP	Ser109 or Thr241	Hippo signaling	enhances the activity of YAP by inhibiting its interaction with LATS1	[9,10]
LATS2	Thr436	Hippo signaling	decreases LATS2 activity by inhibiting MST-mediated phosphorylation	[12]
AMOT-p130	not identified	Hippo signaling	may cause the nuclear accumulation of YAP by decreasing AMOT phosphorylation at Ser175	[11]
LRP6	not identified	Hippo pathway	may diminish LATS activity by decreasing Merlin-LATS interactions	[13]
β-catenin	Thr41	Wnt signaling	increases β-catenin stability by competing with ubiquitinylation-inducing β-catenin phosphorylation	[123,124]
Smad4	Thr63	TGF-β signaling	prevents the GSK3β-mediated proteosomal degradation of Smad4, inducing the TGF-β/SMAD signaling pathway	[68,125]
PKC	not identified	TGF-β signaling	decreases the TGFβRII expression by reducing PKC activity and as a result the TGFβ signaling pathway is reduced	[126,127]
PKA	not identified	GPCR signaling	increases the activity of PKA that bridges the Hippo pathway and GPCR-Gαs signaling	[128,129,130,131]
NOTCH1	not identified	NOTCH signaling	induces the release of the NICD that promotes YAP/TAZ stability	[71,72,73,132]

## Data Availability

The data used to support the findings of this study are available from the corresponding author upon reasonable request.

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
