# Peer review of "O-GlcNAcylation: An Emerging Protein Modification Regulating the Hippo Pathway"

_cancers, 2022, doi:10.3390/cancers14123013_

Round 1

Reviewer 1 Report

In this review the authors discuss the interaction between O-GlcNAcylation and the Hippo pathway. Recent findings suggest that O-GlcNAcylation of core components of the Hippo pathway cause dysregulation, which causes hyperactivated YAP which can in turn further promote OGT expression, setting up a self-perpetuating cycle of dysregulation that can promote tumorigenesis. Further, O-GlcNAc has been implicated in other cellular pathways and regulation/crosstalk with phosphorylation that can also intersect with the Hippo pathway. 

Overall, this was a well thought out, informative and provocative review.  The authors do a good job of piecing together various observations from the literature and presenting ways in which O-GlcNAc and the Hippo pathway and interact.  This review would be of high interest to readers of Cancers.

Author Response

We appreciate your favorable review and we also hope that our paper will be useful to Cancers readers.

Reviewer 2 Report

The Hippo pathway, controlled by a kinase cascade, is a key regulator in the cellular responses. As a result, disruption of the Hippo pathway has been linked to a variety of illnesses, including cancer. The major components of the Hippo signaling pathway include the transcriptional co-activator Yes-associated protein 1 (YAP), nuclear transcription factors (TEAD), and their upstream kinases (MST1/2 and LATS1/2). O-GlcNAcylation is a posttranslational modification (PTM) that regulates the Hippo pathway; it has received much attention in this field. O-GlcNAcylation modulates crosstalk between distinct phosphorylation-mediated signaling pathways. In this article, the authors address the O-GlcNAcylation, essential proteins that regulate the Hippo pathway. This review overviews the Hippo pathway and O-GlcNAcylation and identifies methods through which O-GlcNAcylation produces Hippo pathway dysregulation. The pathogenic importance of the self-perpetuating loop generated by the feedback between O-GlcNAcylation and the Hippo pathway is also discussed by the authors. The figures are informative; however, I have a few concerns about this article. 

I suggest the authors add the table for the O-GlcNAcylation mode of action.

Several basic grammar issues must be improved, and the English style should be edited.

Author Response

We appreciate your constructive suggestions and as you suggested we added a table and made some changes to the grammar and English style.

Reviewer 3 Report

The article by Kim Eunah et al titled as “O-GlcNAcylation: An emerging protein modification regulating the Hippo pathway” elucidates the Hippo-pathway’s intricate regulatory mechanisms, which involve O-GlcN Acylation of the pathway’s candidate molecules. The role of such regulatory mechanisms in the progression of human diseases, including cancers, has been extensively discussed. The article is well-written and well organized. It may be accepted for publication after typos and grammar checks. 

Author Response

We appreciate your favorable review and we have corrected several typos and grammatical errors.

Round 2

Reviewer 2 Report

The authors have improved the revised manuscript. It can be considered for publication.